# Posterior Refinement Improves Sample Efficiency in Bayesian Neural Networks

**Agustinus Kristiadi**
University of Tübingen
agustinus.kristiadi@uni-tuebingen.de

**Runa Eschenhagen**
University of Tübingen
runa.eschenhagen@uni-tuebingen.de

**Philipp Hennig**
University of Tübingen and MPI for Intelligent Systems, Tübingen
philipp.hennig@uni-tuebingen.de

## Abstract

Monte Carlo (MC) integration is the *de facto* method for approximating the predictive distribution of Bayesian neural networks (BNNs). But, even with many MC samples, Gaussian-based BNNs could still yield bad predictive performance due to the posterior approximation's error. Meanwhile, alternatives to MC integration tend to be more expensive and biased. In this work, we experimentally show that the key to good MC-approximated predictive distributions is the quality of the approximate posterior itself. However, previous methods for obtaining accurate posterior approximations are expensive and non-trivial to implement. We, therefore, propose to refine Gaussian approximate posteriors with normalizing flows. When applied to last-layer BNNs, it yields a simple *post hoc* method for improving pre-existing parametric approximations. We show that the resulting posterior approximation is competitive with even the gold-standard full-batch Hamiltonian Monte Carlo.

## 1   Introduction

Predictive uncertainty is crucial in safety-critical systems [1]. Yet, commonly-used neural network (NN) predictive systems are overconfident [2, 3]. Approximate Bayesian inference of NNs, resulting in Bayesian neural networks (BNNs), is a principled way of mitigating this issue [4]. Indeed, even crude approximations of BNNs' posteriors, such as the Laplace approximation [5], can lead to good predictive uncertainty quantification (UQ) performance [6], provided the predictive distributions are correctly computed [7].

A prediction in a BNN amounts to an integration of the likelihood w.r.t. the (approximate) posterior measure. Due to the non-linearity of NNs, no analytic solution to the integral exists, even when the likelihood and the approximate posterior are both Gaussian. A low-cost, unbiased, stochastic approximation can be obtained via Monte Carlo (MC) integration: obtain $S$ samples from the approximate posterior and then compute the empirical expectation of the likelihood w.r.t. these samples. While MC integration is accurate for large $S$, because of the sheer size of modern (B)NNs, virtually all BNNs use small $S$ (typically 10 to 30, see [8–14, etc.]). Due to its well-known error scaling of $\Theta(1/\sqrt{S})$, intuitively, MC integration with a small $S$ is inadequate for an accurate prediction—yet, this has not been studied in depth for BNNs. Furthermore, while linearization of an NN around a point estimate in the parameter space is an alternative to MC integration [7, 15–17], it is generally costly due to the computation of *per-example* Jacobian matrices.

In this work, we study the quality of MC integration for making predictions in BNNs. We show that few-sample MC integration is inaccurate for even an "easy" integral such as when the domain of

36th Conference on Neural Information Processing Systems (NeurIPS 2022).

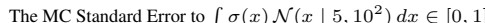

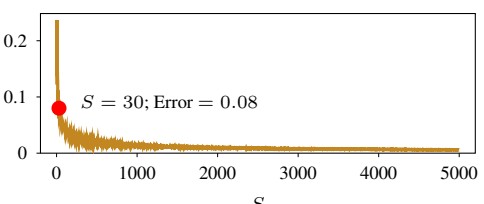

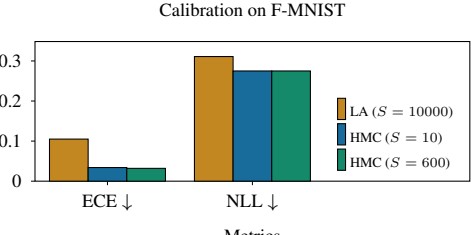

Figure 1: **Left:** Few-sample MC integration is inaccurate for computing classification predictive distribution: the standard error of the MC integration of the logistic-Gaussian integral is large for the commonly used (few) numbers of samples. **Right:** But one can still obtain good predictive performance with a small $S$ if a high-quality posterior approximation, e.g. HMC, is used.

integration is the output space of a binary-classification BNNs, cf. Fig. 1 (left). Further, we show that its alternative, the network linearization, disagrees with large-sample MC integration, making it unsuitable as a general-purpose predictive approximation method, i.e. when the Gauss-Newton matrix is *not* employed.[1] Meanwhile, as indicated by full-batch Markov Chain Monte Carlo methods [18, 19] or even (the Bayesian interpretation of) ensemble methods [20], few-sample MC integration might still be useful for making predictions, provided that the approximate posterior measure is "close enough" to the true posterior—cf. Fig. 1 (right). This implies that one should focus on improving the accuracy of posterior approximations.

Nevertheless, prior methods for obtaining expressive posteriors [e.g., 10, 21] require either significant modification to the NN or storing many copies of the parameters. Moreover, they require training from scratch and thus introduce a significant overhead, which can be undesirable in practical applications. Therefore, we propose a *post hoc* method for "refining" a Gaussian approximate posterior by leveraging normalizing flows [22]. Contrary to the existing normalizing flow methods with *a priori* base distribution (e.g. $\mathcal{N}(0, I)$), the proposed refinement method converges faster with shorter flows, making it cheaper than the naïve application of normalizing flows in BNNs. When used in conjunction with last-layer BNNs, which have been shown to be competitive to their all-layer counterparts [6], the proposed method is simple, cheap, yet competitive to even the gold-standard Hamiltonian Monte Carlo in terms of predictive performance.

To summarize, our contributions are as follows:

(i) We highlight the deficiencies of both few- and many-sample MC integration for computing BNNs' predictive distributions.

(ii) We argue that one must be careful when applying analytic alternatives to MC integration, such as linearization and the probit approximation since they can yield unintended effects.

(iii) We propose a widely-applicable technique for refining parametric posterior approximations by leveraging normalizing flows, which yields a cost-efficient *post hoc* method when used in conjunction with last-layer BNNs.

(iv) We validate the method via extensive experiments and show that refined posteriors are competitive with the much more expensive full-batch Hamiltonian Monte Carlo.

## 2 Preliminaries

### 2.1 Bayesian neural networks

Let $\mathcal{D} := \{(x_i, y_i)\}_{i=1}^m$ be an i.i.d. dataset, sampled from some distributions $p(x)$ on $\mathbb{R}^n$ and $p(y \mid x)$ on $\mathbb{R}^c$. A *Bayesian neural network (BNN)* is a neural network (NN) $f_\theta : \mathbb{R}^n \to \mathbb{R}^c$ with a random parameter $\theta \sim p(\theta)$ on $\mathbb{R}^d$ and a likelihood $p(y \mid f_\theta(x))$, along with the associated posterior $p(\theta \mid \mathcal{D}) \propto p(\theta) \prod_{i=1}^m p(y_i \mid f_\theta(x_i))$. However, the exact posterior $p(\theta \mid \mathcal{D})$ is intractable in general

---

[1]The generalized Gauss-Newton matrix is the *exact* Hessian matrix of a linearized network.

due to its normalization constant. *Approximate BNNs*, which approximate $p(\theta \mid \mathcal{D})$ with only its samples or with a simpler parametric distributions, must thus be employed in most cases.

### 2.1.1 Posterior approximations

The *Laplace approximation* [LA, 5] is one of the simplest approximation methods for BNNs. The main idea is to construct a local Gaussian approximation to $p(\theta \mid \mathcal{D})$, centered at a *maximum a posteriori (MAP)* estimate $\theta_{\text{MAP}} := \arg\max_\theta p(\theta \mid \mathcal{D})$, with the covariance matrix given by the negative inverse-Hessian $(-\nabla_\theta^2 \log p(\theta \mid \mathcal{D})|_{\theta_{\text{MAP}}})^{-1}$. Since the MAP estimation is the standard training procedure for NNs, the LA can be efficiently applied on top of pre-trained NNs [6], especially due to recent efficient second-order optimization libraries [23, 24].

*Variational Bayes* [VB, 25] assumes a family of simple approximate distributions over the parameter space, e.g. $M := \{q(\theta) := \mathcal{N}(\theta \mid \mu, \Sigma) : (\mu, \Sigma) \in \mathbb{R}^{d+d^2}\}$, and finds a $q \in M$ that is the closest to $p(\theta \mid \mathcal{D})$ by minimizing the reverse Kullback-Leibler (KL) divergence $D_{\text{KL}}(q(\theta), p(\theta \mid \mathcal{D}))$ over $M$. Unlike the LA, VB is not constrained to capture just the local mode of the log-posterior landscape. However, VB is not *post hoc* and is generally not as straightforward to implement.

Unlike the previous two approximations, *Markov Chain Monte Carlo (MCMC)* methods do not obtain parametric approximations to the posterior. Instead, they collect samples from asymptotically the true posterior and use them to approximate integrals under the posterior measure. A popular MCMC method is the *Hamiltonian Monte Carlo (HMC)* method [18, 19], where sampling processes are casted as Hamiltonian dynamics. Nevertheless, MCMC methods are generally expensive since they require full batches of data and storage of many copies of the networks' parameters.

### 2.1.2 Predictive approximations

Let $q(\theta)$ be an approximate posterior of a NN $f_\theta$ under $\mathcal{D}$. Given a test point $x_*$, the prediction $y_*$ is distributed as $p(y_* \mid x_*, \mathcal{D}) := \int_{\mathbb{R}^d} p(y_* \mid f_\theta(x_*)) \, q(\theta) \, d\theta$. However, even when both $p(y_* \mid f_\theta(x))$ and $q(\theta)$ are Gaussians, this integral does not have an analytic solution due to the nonlinearity of $f_\theta$. Further approximations on top of the posterior approximation are thus necessary.

**Monte Carlo integration** The *Monte Carlo (MC) integration* is a simple technique to approximate integrals under probability measures. Specifically, in our case,

$$p(y_* \mid x_*, \mathcal{D}) \approx \frac{1}{S} \sum_{s=1}^{S} p(y_* \mid f_{\theta_s}(x_*)); \qquad \text{with } \theta_s \sim q(\theta) \quad \forall s = 1, \dots, S, \tag{1}$$

where $S$ is the number of samples from $q(\theta)$. It is easy to show that MC integration is unbiased with error given by the *standard error* around the mean that scales like $\Theta(1/\sqrt{S})$, see e.g. Murphy [26, Section 2.7.3]. That is, MC integration (slowly) becomes more accurate as the number of samples $S$ increases. In the realm of BNNs, however, $S$ is often chosen to be a small number, e.g. $S = 20$, due to the computational cost of evaluating $f_\theta$.

**Analytic approximations** Since a Gaussian $\mathcal{N}(\theta \mid \mu, \Sigma)$ is the *de facto* choice for $q(\theta)$, an analytic approximation of the marginal output distribution[2] $p(f(x) \mid x, \mathcal{D})$ can be obtained by linearizing $f_\theta$ around $\mu$. Specifically, we perform a first-order Taylor expansion $f_\theta(x) \approx f_\mu(x) + J_f(x) \cdot (\theta - \mu)$ where $J_f(x) \in \mathbb{R}^{c \times d}$ is the Jacobian matrix of the network output w.r.t. $\theta$ at $\mu$ and $\cdot$ denotes matrix multiplication. Under this scheme, denoting $f_* = f(x_*)$, we thus have for each test point $x_*$

$$p(f_* \mid x_*, \mathcal{D}) \approx \mathcal{N}(f_* \mid f_\mu(x_*), J_f(x_*) \Sigma J_f(x_*)^\top) =: \mathcal{N}(f_* \mid f_\mu(x_*), S(x_*)).$$

One can then use this distribution to obtain the predictive distribution $p(y_* \mid x_*, \mathcal{D}) \approx \int_{\mathbb{R}^c} p(y \mid f_*) \, p(f_* \mid x_*, \mathcal{D}) \, df_*$, which again can be approximated via MC integration.

Nevertheless, there exist analytic approximations to this integral [27, 15, 28, 29]. Specifically for binary classification where $p(y \mid f_*) = \sigma(f_*)$ for the logistic function $\sigma$, one can use the so-called *probit approximation* [27, 15]: $p(y = 1 \mid x_*, \mathcal{D}) \approx \sigma(f_\mu(x_*)/\sqrt{1 + \pi/8 \, S(x_*)})$.[3] Moreover, for

---

[2] Where $\theta$ has been marginalized out.

[3] Here, $S(x_*) \in \mathbb{R}_{>0}$ since $f(x_*)$ is a real-valued random variable.

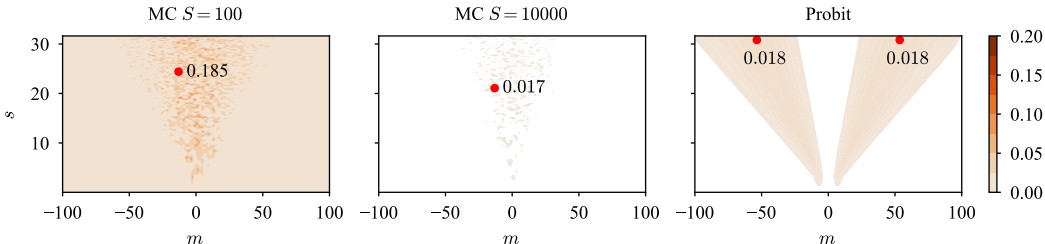

Figure 2: The (absolute) errors of the MC integration and the probit approximation for computing $I(m, s)$ across different values of $m$ and $s$. A trapezoid method with 20000 evaluation points is used as a gold-standard baseline. Red dots indicate the maximum errors. White indicates (near) zero error.

the multi-class case, we can use the *multi-class probit approximation* [MPA, 28]:

$$p(y \mid x_*, \mathcal{D}) \approx \text{softmax}\left(f_\mu(x_*)/\sqrt{1 + \pi/8 \ \text{diag}(S(x_*))}\right), \tag{2}$$

where the vector division is taken component-wise.

## 2.2 Normalizing flows for Bayesian inference

Let $F : \mathbb{R}^d \to \mathbb{R}^d$ be a diffeomorphism[4] in the sense of $C^k(\mathbb{R}^d)$ for a fixed $k \geq 1 \in \mathbb{N}$. If $p$ is a density on $\mathbb{R}^d$,[5] then the density $\widetilde{p}$ of the random variable $\widetilde{\theta} = F(\theta)$ is given by $\widetilde{p}(\widetilde{\theta}) = p(F^{-1}(\widetilde{\theta})) \, |\det J_{F^{-1}}(\widetilde{\theta})|$, where $J_{F^{-1}} = J_F^{-1}$ is the Jacobian matrix of $F^{-1}$. A *normalizing flow (NF)* is a method exploiting this relation [22, 30, 31]. Specifically, it is a way of constructing a sophisticated diffeomorphism $F$, by composing several simple parametric ones. The resulting diffeomorphism thus transforms a simple density into a complicated one in a tractable manner.

Let $F_{\phi_l} : \mathbb{R}^d \to \mathbb{R}^d$ be a diffeomorphism parametrized by $\phi_l \in \mathbb{R}^k$ with a known inverse $F_{\phi_l}^{-1}$ and a known $d \times d$ Jacobian matrix $J_{F_{\phi_l}}$ for $l = 1, \ldots, \ell$. Writing $F_\phi := F_{\phi_\ell} \circ \cdots \circ F_{\phi_1}$ and $\phi := (\phi_l)_{l=1}^\ell$, then the change-of-density formula becomes

$$\widetilde{p}_\phi(\widetilde{\theta}) = p\left(F_\phi^{-1}(\widetilde{\theta})\right) \left| \prod_{l=1}^\ell \det J_{F_{\phi_l}^{-1}}(\widetilde{\theta}) \right|.$$

Given a target posterior density $p^*$, the goal of a normalizing flow is then to estimate $\phi$ s.t. $\widetilde{p}_\phi$ is close to $p^*$. In Bayesian inference, the reverse KL-divergence $D_{\text{KL}}(\widetilde{p}_\phi, p^*)$ is often used.

# 3 Pitfalls of BNNs' Approximate Predictive Distributions

In this section, we study the failure modes of BNNs, especially last-layer ones, for predictive uncertainty calibration. We shall show that there are two parts contributing to the inaccuracy in the predictive distribution of a BNN: (i) the weight-space approximation and (ii) the integration method to do Bayesian model averaging. We shall observe that while the accuracy of the integration method is impactful (i.e., the number of samples in MC integration), it is less important than the quality of the posterior approximation itself. That is, higher-quality weight-space approximations—even in last-layer BNNs—yield samples that are more efficient: Few-sample MC integration can already yield good predictive performance in this case.

## 3.1 Accurate MC integration requires many samples

We begin our analysis with the simplest yet practically relevant case. Let $x_* \in \mathbb{R}^n$ and $f_* := f(x_*)$, with $p(f_* \mid x_*, \mathcal{D}) = \mathcal{N}(f_* \mid m, s^2)$ for some $m \in \mathbb{R}$ and $s^2 \in \mathbb{R}_{>0}$. We are interested in computing

---

[4]A smooth function with a smooth inverse.

[5]All densities considered in this paper are assumed to be w.r.t. the Lebesgue measure.

the integral (cf. Fig. 3)

$$I(m, s) := p(y \mid x_*, \mathcal{D}) = \int_{\mathbb{R}} \sigma(f_*) \mathcal{N}(f_* \mid m, s^2) \, df_*.$$

Note that this integral is prevalent in practice, e.g. in linearized or last-layer classification BNNs [4, 7, 17, 32].

We compare MC integration with $S = 100$ against the probit approximation, using a trapezoid quadrature with 20000 evaluation points to represent a gold-standard baseline. That is, we compute the discrepancy $|\widetilde{I}(m, s) - I_*(m, s)|$ where $\widetilde{I}(m, s)$ is $I(m, s)$ computed either with MC integration or the probit approximation, and $I_*(m, s)$ obtained via the trapezoid method.

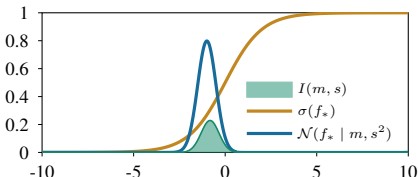

Figure 3: The intuition of $I(m, s)$.

The results are in Fig. 2: Even with $S = 100$ samples—larger than the usual $S = 10\text{-}30$—MC integration is inaccurate. Its error can be as high as 0.18, which is substantial considering $I(m, s) \in [0, 1]$. As $S$ increases beyond 10000, MC integration improves and eventually overtakes the probit approximation, as to be expected given its theoretical guarantees. This highlights the flaw of few-sample MC integration—accurate MC integration generally requires a large number of samples.

### 3.2 Many-sample MC integration is not sufficient

However, even with a large $S$, MC integration can still fail to yield good predictive performance in BNNs. This can happen when the approximation $q(\theta)$ used in (1) is an inaccurate approximation of $p(\theta \mid \mathcal{D})$—virtually the case for every parametric BNN. Thankfully, there is some evidence that the error of MC integration might be relatively small in comparison to the error generated by crude posterior approximations. As an extreme example, Deep Ensemble [20] and its variants, even though they perform MC integration with a small number of samples (usually $S = 5$), generally yield better approximations to the predictive distributions. This is perhaps due to their multimodality, i.e. due to their finer-grained posterior approximations.

Table 1: The expected calibration error (ECE) and negative log-likelihood (NLL) of LA ($S = 10000$) and HMC ($S = 10$).

| Methods | ECE ↓ | NLL ↓ |
|---|---|---|
| **F-MNIST** | | |
| LA | 10.5±0.4 | 0.311±0.005 |
| HMC | **3.4**±0.2 | **0.275**±0.004 |
| **CIFAR-10** | | |
| LA | 4.9±0.2 | 0.161±0.001 |
| HMC | **4.2**±0.2 | **0.158**±0.001 |

To show this more concretely, consider the following experiment. We take Fashion-MNIST (F-MNIST) pre-trained LeNet and CIFAR-10 pre-trained WideResNet-16-4. For each case, we perform a last-layer Laplace approximation with the exact Hessian and a full-batch NUTS-HMC [19], under the same prior and likelihood. We find in Table 1 that HMC, even with few samples, yields better-calibrated predictive distribution than the LA with three orders of magnitude more samples. This finding validates the widely-believed wisdoms [9, 10, 14, 32, etc.] that highly accurate posterior approximations are most important for BNNs. Furthermore, this also shows that with a fine-grained posterior approximation, the predictive performance of a BNN is less sensitive to the number of MC samples, leading to better test-time efficiency.

### 3.3 Analytic alternatives to MC integration are not the definitive answers

Of course, fine-grained posterior approximations are expensive. So, a natural question is whether one can keep a cheap but crude posterior approximation by replacing MC integration. Network linearization seems to be a prime candidate to replace MC integration in the case of Gaussian approximations [7, 15]. But it poses several problems: First, it requires relatively expensive computation of the per-example Jacobian $J_f(x_*)$, where *for each* test point $x_*$, one must store the associated $c \times d$ matrix, where $d$ could easily be tens of millions (e.g. in ResNets) and $c$ could be in the order of thousands (e.g. ImageNet). Second, while Immer et al. [7] argued that network linearization is the correct way to make predictions in Gauss-Newton-based Gaussian approximations, it is not generally applicable. We show this in Fig. 4: Everything else being equal, MC integration ($S = 10000$) and linearization yield different results especially in terms of predictive uncertainty. Since one should

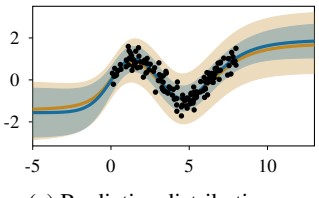

(a) Predictive distributions.

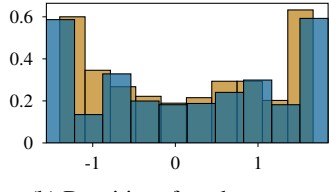

(b) Densities of pred. means.

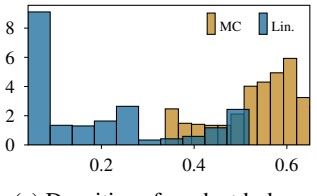

(c) Densities of pred. std. dev.

Figure 4: Predictive distributions, computed via MC integration ($S = 10000$) and network linearization, of a BNN with a weight-space Gaussian approximate posterior (an all-layer LA with the exact Hessian on a two-layer NN under a toy regression dataset).

prefer MC integration in this many-sample regime, linearization is thus not accurate for the general cases.

Moreover, we show that the multi-class probit approximation (MPA), which is often used on top of a linearized output distribution $p(f(x_*) \mid x_*, \mathcal{D})$ to obtain the predictive distribution $p(y_* \mid x_*, \mathcal{D})$ [6, 17, 32], can also obscure the predictive performance, albeit often for the better [6, 32]. To that end, we use the same experiment setting as Section 3.2 and Table 1, and compare the MPA against MC integration. The results are in Table 2.

Indeed, we see that the MPA yields better-calibrated predictive distributions. However, this result should be taken with a grain of salt: The MPA ignores the off-diagonal elements of the covariance of $p(f(x_*) \mid x_*, \mathcal{D})$, as shown in

Table 2: Calibration of MC integration $S = 10000$ and the multi-class probit approximation.

| Methods | ECE $\downarrow$ | NLL $\downarrow$ |
|---|---|---|
| **F-MNIST** | | |
| MPA | **3.3**$\pm$0.2 | **0.281**$\pm$0.002 |
| MC | 10.5$\pm$0.4 | 0.311$\pm$0.005 |
| **CIFAR-10** | | |
| MPA | **3.8**$\pm$0.1 | **0.161**$\pm$0.001 |
| MC | 4.9$\pm$0.2 | **0.161**$\pm$0.001 |

(2)—further detail is in Appendix A. That is, the MPA "biases" the predictive distribution $p(y \mid x_*, \mathcal{D})$ since it generally assumes that $f_*$ has lower uncertainty than it actually has. And since $p(f_* \mid x_*, \mathcal{D})$ is usually induced by the LA or VB which are often underconfident on large networks [7, 33, 34], the bias of MPA towards overconfidence thus counterbalances the underconfidence of $p(f_* \mid x_*, \mathcal{D})$. Therefore, in this case, the MPA can yield better-calibrated predictive distributions than MC integration [32, 6]. Nevertheless, it can also fail even in simple cases such as in Fig. 5,[6] where the MPA yields underconfident predictions even near the training data (notice the lighter shade in Fig. 5, bottom). Moreover, the structured nature of the error of the MPA (cf. Fig. 2 for the binary case) might also contribute to the cases where the MPA differs from MC integration. Both examples above thus highlights the need for careful consideration when analytic alternatives to MC integration are employed.

## 4 Refining Gaussian Approximate Posteriors

The previous analysis indicates the importance of accurate approximate posteriors $q(\theta)$ for BNNs' predictive distributions. However, the current *de facto* way of obtaining accurate posterior approximations, HMC [18, 19], are *very* expensive since they require a full-batch of data in their updates [35]. While mini-batch versions of HMC exist, they do not seem to yield as good of results as their full-batch counterparts. Indeed, Daxberger et al. [6] showed that a well-tuned *last-layer* LA can outperform a state-of-the-art *all-layer* stochastic-gradient MCMC method [21]. Furthermore, these sample-based methods—both the full- and mini-batch versions, along with deep ensembles and their variants—are costly in terms of storage since one effectively must store $S$ copies of the network's parameters. We, therefore, propose a simple *post hoc* technique for "refining"

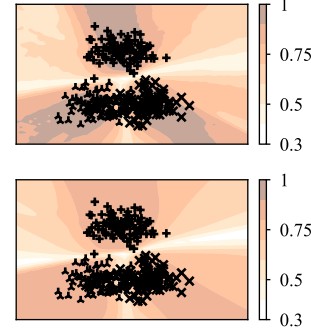

Figure 5: Confidence estimate of MC integration (**top**) and the MPA (**bottom**).

---

[6]An all-layer full-Hessian LA on a three-layer `tanh` network is used.

Gaussian approximations using normalizing flows. The resulting method is thus analytic yet can produce high-quality samples.

Let $f_\theta$ be a NN equipped with a Gaussian approximate posterior $q(\theta) = \mathcal{N}(\theta \mid \mu, \Sigma)$ (e.g., via a LA, VB, or SWAG [13]) on the parameter space under a dataset $\mathcal{D}$. Given a NF $F_\phi$ of length $\ell$ with parameter $\phi$, we obtain the *refined posterior* by

$$\widetilde{q}_\phi(\widetilde{\theta}) = q(F_\phi^{-1}(\widetilde{\theta})) \left| \det J_{F_\phi}(\widetilde{\theta}) \right|^{-1}. \tag{3}$$

Then, a *refinement* of $q(\theta)$ amounts to minimizing the reverse KL-divergence to the true posterior, using the evidence lower bound (ELBO) as a proxy ($\mathbb{H}$ below is the entropy functional):

$$\phi^* = \arg\max_\phi \; \mathop{\mathbb{E}}_{\widetilde{\theta} \sim \widetilde{q}_\phi} \left[ \log p(\mathcal{D} \mid f_{\widehat{\theta}}) + \log p(\widetilde{\theta}) \right] + \mathbb{H}\left[\widetilde{q}_\phi\right], \tag{4}$$

Given a refined posterior $q_{\phi^*}(\widetilde{\theta})$ and a test point $x^*$, we can obtain the predictive distribution via MC integration:

$$p(y^* \mid x^*, \mathcal{D}) \approx \frac{1}{S} \sum_{s=1}^{S} p\left(y^* \mid f_{\widetilde{\theta}_s}(x^*)\right); \quad \text{where} \;\; \widetilde{\theta}_s = F_{\phi^*}(\theta_s); \;\; \theta_s \sim q(\theta) \quad \forall s = 1, \dots, S.$$

Due to the expressiveness of NFs [36], we can expect based on the previous analysis that a large $S$ is not necessary here to obtain good predictive performance. We shall validate this in Section 6. Last but not least, this refinement technique is especially useful for last-layer BNNs since their parameter spaces typically have manageable dimensions, cf. Section 6.4.[7]

## 5 Related Work

Normalizing flows have previously been used for approximate Bayesian inference in BNNs. An obvious way to do so, based on the flexibility of NFs in approximating any density [36], would be to apply a NF on top of the standard normal distribution $\mathcal{N}(\theta \mid 0, I)$ to approximate $p(\theta \mid \mathcal{D})$, see e.g. Izmailov et al. [38] and the default implementation of variational approximation with NF in Pyro [39]. We show in Section 6.3 that the subtle difference that we make—using an approximate posterior instead of an *a priori* distribution—is more cost-effective. In a more sophisticated model, Louizos and Welling [10] combine VB with NF by assuming a compound distribution on each NN's weight matrix and using the NF to obtain an expressive mixing distribution. However, their method requires training both the BNN and the NF jointly from scratch. In an adjacent field, Maroñas et al. [40] use NFs to transform Gaussian process priors.

Posterior refinement in approximate Bayesian inference has recently been studied. Immer et al. [7] proposes to refine the LA using a Gaussian-based VB and Gaussian processes. However, this implies that they still assume a Gaussian posterior. To obtain a non-Gaussian posterior, Miller et al. [41] form a mixture-of-Gaussians approximation by iteratively adding component distributions. But, at *every* iteration, their methods require a full ELBO optimization, making it costly for BNNs. A lower-cost LA-based alternative to their work has also been proposed by Eschenhagen et al. [32]. These methods have high storage costs since they must store many high-dimensional Gaussians. By contrast, our method only does an ELBO optimization once and only requires the storage of the base Gaussian and the parameters of a NF. In a similar vein, Havasi et al. [42] perform an iterative ELBO optimization for refining a sample of a variational approximation. But, this inner-loop optimization needs to be conducted each time a sample is drawn, e.g., during MC integration at test time. Our method, on the other hand, only requires sampling from a Gaussian and evaluations of a NF, cf. (4).

While our results also reaffirm the widely held belief that single-mode Gaussian approximations are inferior to more fine-grained counterparts, our conclusion differs slightly from that of Wilson and Izmailov [43]: They argue "[...] Ultimately, the goal is to accurately compute the predictive distribution, rather than find a generally accurate representation of the posterior. [...]" and subsequently propose a multi-modal approximation to the true posterior. Here, we show that finding an accurate (non-Gaussian) weight-space posterior approximation can still be a worthy goal, even when only utilizing a single mode of the true posterior.

---

[7]E.g., WideResNets' last-layer features' dimensionality typically range from 256 to 2048 [37].

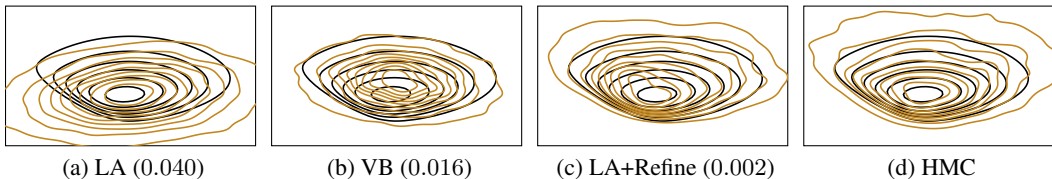

| (a) LA (0.040) | (b) VB (0.016) | (c) LA+Refine (0.002) | (d) HMC |

Figure 6: Comparison of approximate posterior densities, visualized via a kernel density estimation, on a 2D logistic regression problem. **Black contour:** The exact posterior contour, up to a normalizing constant. **Colored contour:** The kernel density estimate obtained from the posterior samples of each method. **Number:** The MMD distance to HMC's samples; lower is better.

## 6 Experiments

Code available at: `https://github.com/runame/laplace-refinement`. See Appendix B and Appendix C for additional details.

### 6.1 Setups

**Datasets** We validate our method using standard classification datasets: Fashion-MNIST (F-MNIST), CIFAR-10, and CIFAR-100.[8] For the out-of-distribution (OOD) detection task, we use three standard OOD test sets for each in-distribution dataset, see Appendix B.2 for the full list. Finally, for the toy logistic regression experiment, a dataset of size 50 is generated by sampling from a bivariate, bimodal Gaussian.

**Network architectures** For the F-MNIST experiments, we use the LeNet-5 architecture [44]. Meanwhile, for the CIFAR experiments, we use the WideResNet architecture with a depth of 16 and widen factor of 4 [WRN-16-4, 37]. For the NF, we use the radial flow [22] which is among the simplest and cheapest non-trivial NF architectures.

**Baselines** We focus on last-layer Bayesian methods to validate the refinement technique.[9] We use the LA to obtain the base distribution for our refinement method—following recent practice [6], we tune the prior precision via *post hoc* marginal likelihood maximization. The No-U-Turn-Sampler Hamiltonian Monte Carlo [HMC, 18, 19] with 600 samples is used as a gold-standard baseline—all HMC results presented in this paper are well-converged in term of the Gelman-Rubin diagnostic [45], see Appendix B. Furthermore, we compare our method against recent, all-layer BNN baselines: variational Bayes with the Flipout estimator [VB, 46] and cyclical stochastic-gradient HMC [CSGHMC, 21]. For all methods, we use MC integration with 20 samples to obtain the predictive distribution, except for HMC and CSGHMC where we use $S = 600$ and $S = 12$, respectively. We use prior precisions of 510 and 40 for the last-layer F-MNIST and CIFAR experiments, respectively. These prior precisions are obtained via grid search on the respective HMC baseline, maximizing validation log-likelihood. Additionally, we use MAP and its temperature-scaled version (MAP-Temp) as non-Bayesian baselines. More implementation details are in Appendix B.

**Metrics** We use standard metrics to measure both calibration and OOD-detection performance. For the former, we use the negative log-likelihood (NLL) and the expected calibration error [ECE, 47]. For the latter, we employ the false-positive rate at $95\%$ true-positive rate (FPR95). Finally, to measure the closeness between an approximate posterior to the true posterior, we use the maximum-mean discrepancy [MMD, 48] distance between the said approximation's samples to the HMC samples, i.e. we use HMC as a proxy to the true posterior.

### 6.2 Toy example

We visualize different approximations to the posterior of the toy logistic regression problem in Fig. 6. Note that the true posterior density is non-symmetric and non-Gaussian.

---

[8]Additionally, see Appendix C for results on text classifications.

[9]Results for an all-layer network are in Table 6 (Appendix C).

Table 3: In-distribution calibration performance. All-layer baselines are marked with an asterisk.

| Methods | F-MNIST MMD ↓ | F-MNIST NLL ↓ | CIFAR-10 MMD ↓ | CIFAR-10 NLL ↓ | CIFAR-100 MMD ↓ | CIFAR-100 NLL ↓ |
|---|---|---|---|---|---|---|
| MAP | 1.093±0.003 | 0.3116±0.0049 | 0.438±0.001 | 0.1698±0.0009 | 0.416±0.001 | 0.9365±0.0063 |
| MAP-Temp | 1.093±0.003 | 0.2694±0.0025 | 0.438±0.001 | 0.1545±0.0007 | 0.416±0.001 | 0.9155±0.0047 |
| VB* | - | 0.2673±0.0016 | - | 0.2823±0.0020 | - | 1.2638±0.0059 |
| CSGHMC* | - | 0.2854±0.0018 | - | 0.2101±0.0033 | - | 0.9892±0.0070 |
| LA | 0.418±0.002 | 0.3076±0.0046 | 0.299±0.001 | 0.1672±0.0009 | 0.063±0.000 | 0.9865±0.0057 |
| LA-Refine-1 | 0.356±0.004 | 0.2752±0.0031 | 0.346±0.000 | 0.1616±0.0007 | 0.063±0.000 | 0.9548±0.0062 |
| LA-Refine-5 | 0.022±0.002 | 0.2699±0.0028 | 0.290±0.000 | 0.1582±0.0007 | 0.018±0.000 | 0.9073±0.0062 |
| LA-Refine-10 | 0.013±0.002 | 0.2701±0.0028 | 0.130±0.001 | 0.1577±0.0008 | 0.019±0.000 | 0.9037±0.0058 |
| LA-Refine-30 | 0.012±0.002 | 0.2701±0.0028 | 0.002±0.000 | 0.1581±0.0008 | 0.020±0.000 | 0.9035±0.0055 |
| HMC | 0.000±0.000 | 0.2699±0.0028 | 0.000±0.000 | 0.1581±0.0008 | 0.000±0.000 | 0.8849±0.0047 |

The LA matches the weight-space posterior mean, but is inaccurate the further away from it. It even assigns probability mass on what are supposed to be low-density regions. While VB yields a more accurate result than the LA, it still assigns some probability mass on low-density regions due to the symmetry of the Gaussian approximation. Furthermore, it is unable to match the posterior's mode well. The proposed refinement method, on the other hand, is able to make the LA more accurate—it yields a skewed, non-Gaussian approximation, similar to HMC.

We further quantify the previous observation using the MMD distance between each approximation's samples and HMC's samples. The LA, as expected, obtain the worst weight-space MMD. While VB is better than the LA with an MMD, the refined LAs achieve even better MMDs. This quantifies the previous visual observation.

## 6.3 Image classification

We present the calibration results in Table 3 using the LA and HMC as baselines, which represent two "extremes" in the continuum of posterior approximations. As has previously shown by e.g. Guo et al. [49], the vanilla MAP approximation yields uncalibrated, low-quality predictive distributions in terms of NLL and ECE. While the LA is a cheap way to improve MAP predictive, its predictive performance is still lagging behind HMC. By refining it with a NF, the LA becomes even better and closer to the HMC predictive. We also note that one does not need a complicated or long (thus expensive) NF to achieve these improvements. Furthermore, we observe a positive correlation between posterior-approximation quality (measured via MMD) and predictive quality. Considering that $S = 20$ is used, this validates our hypothesis that one can "get away" with fewer MC samples when accurate weight-space posterior approximations are employed.

Moreover, we present out-of-distribution (OOD) data detection in Table 4. We observe that while the last-layer LA baseline can already be better than all-layer baselines—as also observed by Daxberger et al. [6]—refining it can yield better results: Even with a small NF, e.g., $\ell = 5$, the OOD detection performance of the refined LA is close to that of the gold-standard HMC.

Finally, we show that it is indeed desirable to do *refinement*, i.e. using an approximate posterior as the base distribution of the NF, instead of starting from scratch, i.e. starting from a data-independent distribution such as $\mathcal{N}(0, I)$. As shown in Fig. 7, starting from an approximate posterior

Table 4: OOD detection in terms of FPR95 (in percent, lower is better), averaged over three test sets and five seeds. All-layer baselines are asterisk-marked.

| Methods | CIFAR-10 | CIFAR-100 |
|---|---|---|
| VB* | 62.9±2.0 | 80.8±1.0 |
| CSGHMC* | 58.7±1.6 | 79.3±1.0 |
| LA | 49.2±2.4 | 79.6±1.0 |
| LA-Refine-1 | 47.7±2.1 | 77.3±0.7 |
| LA-Refine-5 | 46.8±2.2 | 77.8±0.7 |
| LA-Refine-10 | 46.2±2.3 | 77.8±0.7 |
| LA-Refine-30 | 46.1±2.3 | 77.9±0.8 |
| HMC | 46.0±2.3 | 77.8±0.9 |

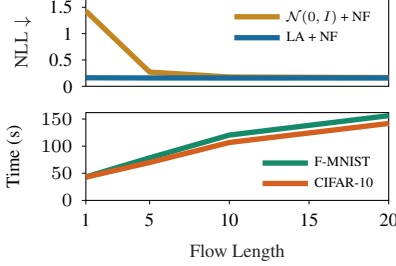

Figure 7: Calibration and wall-clock refinement time vs. flow length.

yields better predictive distributions faster than when $\mathcal{N}(0, I)$ is used as the base distribution of the NF. This is particularly important since the computational cost of a NF depends on its length: We see a $53\%$ increase in training time from $\ell = 5$ to $\ell = 10$—the latter is required for the *a priori* NF approximation to yield similar predictive performance to the refined posterior.

### 6.4 Costs

The proposed refinement technique is *post hoc* and cheap when applied to last-layer BNNs. Suppose one already has a last-layer Gaussian approximate posterior. Using a standard consumer GPU (Nvidia RTX 2080Ti), each epoch a length-5 NF's optimization takes around $3.4$ seconds. From our experiments, we found that a low number of epochs (we use 20) is already sufficient for improving a crude approximate posterior. Thus, the entire refinement process is quick, especially when compared to MAP estimation, ELBO optimization, or HMC sampling. Additional details in Appendix C.3.

## 7 Concluding Remarks

### 7.1 Limitations

A NF is a composition of diffeomorphisms—it is bijective and invertible. Since we use the NF to transform the density on the parameter space of a NN, which has tens of millions of dimensions, the proposed refinement framework is thus costly for *all-layer* BNNs or when the number of last-layer parameters is prohibitively large. Interesting future works to alleviate this limitation are to refine a posterior in the subspace of the parameter space [38] or in the "pruned" parameter space [50].

Moreover, by refining a Gaussian with a NF, one loses the convenient properties associated with Gaussians, making the resulting BNN unsuitable for cheaply addressing tasks such as continual learning [51] and model selection [52]. Nevertheless, this is not unique to our method—any sample-based method such as MCMC and ensembles are unsuitable for such tasks.

### 7.2 Conclusion

We have shown that MC integration tends to produce underperforming predictive distributions under a crude posterior approximation, even with a large number of samples. While recently analytic approximations such as network linearization and the probit approximation have been extensively used in BNNs, we show that one must be careful with them since they are biased and do not have theoretical guarantees like MC integration. Inspired by the high predictive performance of full-batch HMC—even when a low number of samples are used—we confirm the widely-held belief that a finer-grained posterior approximation is a key to achieving better predictive performance in BNNs. To that end, we proposed a *post hoc* method for last-layer parametric BNNs, based on normalizing flows, that can cheaply refine crude posterior approximations. In conjunction with the *post hoc* last-layer Laplace approximation, the proposed method can thus give practitioners similar predictive performance to full-batch, expensive HMC in a cost-efficient manner.

## Acknowledgments and Disclosure of Funding

The authors gratefully acknowledge financial support by the European Research Council through ERC StG Action 757275 / PANAMA; the DFG Cluster of Excellence "Machine Learning - New Perspectives for Science", EXC 2064/1, project number 390727645; the German Federal Ministry of Education and Research (BMBF) through the Tübingen AI Center (FKZ: 01IS18039A); and funds from the Ministry of Science, Research and Arts of the State of Baden-Württemberg. The authors are also grateful to Felix Dangel for feedback. AK is also grateful to the International Max Planck Research School for Intelligent Systems (IMPRS-IS) for support.

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
