# Posterior Refinement Improves Sample Efficiency in Bayesian Neural Networks

## Appendix A    Derivation of the Multi-Class Probit Approximation

**Remark.**    This derivation first appeared in the first author's blog post [53].

Let $p(z) := \mathcal{N}(z \mid \mu, \Sigma)$ be the distribution a random variable $z \in \mathbb{R}^c$. The multi-class probit approximation (MPA) is defined by

$$\int_{\mathbb{R}^c} \mathrm{softmax}(z)\, p(z)\, dz \approx \mathrm{softmax}\left( \frac{\mu}{\sqrt{1 + \pi/8\, \mathrm{diag}(\Sigma)}} \right),$$

where the vector division is defined component-wise. Its derivation, based on Lu et al. [54] is as follows.

Notice that we can write the $i$-th component of $\mathrm{softmax}(z)$ as $1/(1 + \sum_{j \neq i} \exp(-(z_i - z_j)))$. So, for each $i = 1, \dots, c$, using $z_{ij} := z_i - z_j$, we can write

$$\frac{1}{1 + \sum_{j \neq i} \exp(-z_{ij})} = \frac{1}{1 - (K-1) + \sum_{j \neq i} \frac{1}{\frac{1}{1 + \exp(-z_{ij})}}}$$

$$= \frac{1}{2 - K + \sum_{j \neq i} \frac{1}{\sigma(z_{ij})}}.$$

Then, we use the following approximations:

1. $\mathbb{E}(f(x)) \approx f(\mathbb{E}(x))$ for a non-zero function $f$,
2. the mean-field approximation $\mathcal{N}(z \mid \mu, \Sigma) \approx \mathcal{N}(z \mid \mu, \mathrm{diag}(\Sigma))$, and thus we have $z_{ij} := z_i - z_j \sim \mathcal{N}(z_{ij} \mid \mu_i - \mu_j, \Sigma_{ii} + \Sigma_{jj})$, and
3. using the (binary) probit approximation (see Section 2.1.2), with a further approximation:

$$\int_{\mathbb{R}} \sigma(z_{ij}) \mathcal{N}(z_{ij} \mid \mu_i - \mu_j, \Sigma_{ii} + \Sigma_{jj})\, dz_{ij} \approx \sigma\left( \frac{\mu_i - \mu_j}{\sqrt{1 + \pi/8\, \Sigma_{ii} + \Sigma_{jj}}} \right)$$

$$\approx \sigma\left( \frac{\mu_i}{\sqrt{1 + \pi/8\, \Sigma_{ii}}} - \frac{\mu_j}{\sqrt{1 + \pi/8\, \Sigma_{jj}}} \right),$$

we obtain

$$\int_{\mathbb{R}^c} \mathrm{softmax}_i(z)\, \mathcal{N}(z \mid \mu, \Sigma) \approx \frac{1}{2 - K + \sum_{j \neq i} \frac{1}{\mathbb{E}\, \sigma(z_{ij})}}$$

$$\approx \frac{1}{2 - K + \sum_{j \neq i} \frac{1}{\sigma\left( \frac{\mu_i}{\sqrt{1 + \pi/8\, \Sigma_{ii}}} - \frac{\mu_j}{\sqrt{1 + \pi/8\, \Sigma_{jj}}} \right)}}$$

$$= \frac{1}{1 + \sum_{j \neq i} \exp\left( -\left( \frac{\mu_i}{\sqrt{1 + \pi/8\, \Sigma_{ii}}} - \frac{\mu_j}{\sqrt{1 + \pi/8\, \Sigma_{jj}}} \right) \right)}$$

$$= \frac{\exp\left( \mu_i / \sqrt{1 + \pi/8\, \Sigma_{ii}} \right)}{\sum_{j=1}^{k} \exp\left( \mu_j / \sqrt{1 + \pi/8\, \Sigma_{jj}} \right)}$$

We identify the last equation above as the $i$-th component of $\mathrm{softmax}\left( \frac{\mu}{\sqrt{1 + \pi/8\, \mathrm{diag}(\Sigma)}} \right)$.

# Appendix B    Implementation Details

## B.1    Training

We use the Pyro library [39] to implement the normalizing flow (NF) used for the refinement. The NF is trained by maximizing the evidence lower bound using the Adam optimizer [55] and the cosine learning rate decay [56] for 20 epochs, with an initial learning rate of $0.001$. Following [35], we do not use data augmentation.

For the HMC baseline, we use the default implementation of NUTS in Pyro. We confirm that the HMC used in our experiments are well-converged: The average Gelman-Rubin $\widehat{R}$'s are $0.998$, $0.999$, $0.997$, and $1.096$—below the standard threshold of $1.1$—for the last-layer F-MNIST, last-layer CIFAR-10, last-layer CIFAR-100, and all-layer F-MNIST experiments, respectively.

For the MAP, VB, and CSGHMC baselines, we use the same settings as Daxberger et al. [6]: We train them for $100$ epochs with an initial learning rate of $0.1$, annealed via the cosine decay method [56]. The minibatch size is $128$, and data augmentation is employed. For MAP, we use weight decay of $5 \times 10^{-4}$. For VB and CSGHMC, we use the prior precision corresponding to the previous weight decay value.

For the LA baseline, we use the `laplace-torch` library [6]. The diagonal Hessian is used for CIFAR-100 and all-layer F-MNIST, while the full Hessian is used for other cases. Following the current best-practice in LA, we tune the prior precision with *post hoc* marginal likelihood maximization [6].

Finally, for methods which require validation data, e.g. HMC (for finding the optimal prior precision), we obtain a validation test set of size $2000$ by randomly splitting a test set. Note that, these validation data are not used for testing.

## B.2    Datasets

For the dataset-shift experiment, we use the following test sets: Rotated F-MNIST and Corrupted CIFAR-10 [57, 58]. Meanwhile, we use the following OOD test sets for each the in-distribution training set:

- **F-MNIST:** MNIST, K-MNIST, E-MNIST.
- **CIFAR-10:** LSUN, SVHN, CIFAR-100.
- **CIFAR-100:** LSUN, SVHN, CIFAR-10.

# Appendix C    Additional Results

## C.1    Image classification

To complement Table 3 in the main text, we present results for additional metrics (accuracy, Brier score, and ECE) in Table 5. We see that the trend Table 3 is also observable here. We also show that the refinem In Table 6, we observe that refining an *all-layer* posterior improves its predictive quality further.[10]

In Table 7, we present the detailed, non-averaged results to complement Table 4. Moreover, we also present dataset-shift results on standard benchmark problems (Rotated F-MNIST and Corrupted CIFAR-10). In both cases, we observe that the performance of the refined posterior approaches HMC's.

## C.2    Text classification

We further validate the proposed method on text classification problems. We the benchmark in Hendrycks et al. [59], Kristiadi et al. [60]: A GRU recurrent network architecture [61] is used and trained for 10 epochs under the 20NG, SST, and TREC datasets. The prior precision for HMC and the refinement method is $40$. We use the same normalizing flow as in the image classification experiment and follow a similar training procedure.

---

[10]The network is a two-layer fully-connected ReLU network with $50$ hidden units.

Table 5: In-distribution calibration performance.

| Methods | F-MNIST | | | CIFAR-10 | | | CIFAR-100 | | |
|---|---|---|---|---|---|---|---|---|---|
| | Acc. ↑ | Brier ↓ | ECE ↓ | Acc. ↑ | Brier ↓ | ECE ↓ | Acc. ↑ | Brier ↓ | ECE ↓ |
| MAP | 90.4±0.1 | 0.1445±0.0008 | 11.7±0.3 | 94.8±0.1 | 0.0790±0.0004 | 10.5±0.2 | 76.5±0.1 | 0.3396±0.0012 | 13.7±0.2 |
| MAP-Temp | 90.4±0.1 | 0.1377±0.0009 | 3.2±0.0 | 94.9±0.1 | 0.0767±0.0008 | 6.0±1.3 | 75.9±0.1 | 0.3394±0.0007 | 8.9±0.2 |
| VB* | 90.5±0.1 | 0.1377±0.0013 | 3.9±0.2 | 91.1±0.1 | 0.1333±0.0008 | 5.3±0.2 | 69.6±0.2 | 0.4144±0.0016 | 5.5±0.2 |
| CSGHMC* | 89.6±0.1 | 0.1492±0.0011 | 3.7±0.1 | 93.6±0.2 | 0.0975±0.0017 | 7.2±0.3 | 73.8±0.1 | 0.3647±0.0019 | 7.1±0.1 |
| LA | 90.4±0.0 | 0.1439±0.0008 | 11.1±0.2 | 94.8±0.0 | 0.0785±0.0004 | 9.8±0.3 | 75.6±0.1 | 0.3529±0.0009 | 9.7±0.1 |
| LA-Refine-1 | 90.4±0.1 | 0.1386±0.0007 | 5.2±0.2 | 94.7±0.0 | 0.0776±0.0003 | 4.3±0.2 | 75.9±0.1 | 0.3445±0.0010 | 8.0±0.2 |
| LA-Refine-5 | 90.4±0.1 | 0.1375±0.0009 | 3.2±0.1 | 94.8±0.1 | 0.0768±0.0004 | 4.3±0.2 | 76.2±0.1 | 0.3311±0.0007 | 4.5±0.2 |
| LA-Refine-10 | 90.5±0.1 | 0.1376±0.0008 | 3.6±0.1 | 94.9±0.1 | 0.0765±0.0004 | 4.4±0.2 | 76.1±0.1 | 0.3312±0.0008 | 4.4±0.1 |
| LA-Refine-30 | 90.4±0.1 | 0.1376±0.0009 | 3.5±0.1 | 94.9±0.1 | 0.0765±0.0004 | 4.4±0.1 | 76.1±0.1 | 0.3315±0.0007 | 4.2±0.2 |
| HMC | 90.4±0.1 | 0.1375±0.0009 | 3.4±0.0 | 94.9±0.1 | 0.0765±0.0004 | 4.3±0.1 | 76.4±0.1 | 0.3283±0.0007 | 4.6±0.1 |

Table 6: Calibration of all-layer BNNs on F-MNIST. The architecture is two-layer ReLU fully-connected network with 50 hidden units.

| Methods | MMD ↓ | Acc. ↑ | NLL ↓ | Brier ↓ | ECE ↓ |
|---|---|---|---|---|---|
| LA | 0.278±0.003 | 88.0±0.1 | 0.3597±0.0009 | 0.18±0.0006 | 7.7±0.1 |
| LA-Refine-1 | 0.194±0.006 | 87.6±0.1 | 0.3564±0.0015 | 0.1807±0.0006 | 6.1±0.1 |
| LA-Refine-5 | 0.19±0.006 | 87.6±0.1 | 0.3483±0.0012 | 0.1781±0.0005 | 4.9±0.3 |
| LA-Refine-10 | 0.186±0.006 | 87.7±0.1 | 0.3459±0.0008 | 0.1771±0.0004 | 4.7±0.3 |
| LA-Refine-30 | 0.183±0.006 | 87.8±0.1 | 0.3432±0.0014 | 0.176±0.0007 | 4.6±0.3 |
| HMC | - | 89.7±0.0 | 0.2908±0.0002 | 0.1502±0.0001 | 4.5±0.1 |

The results are in Table 8. We observe similar results as in the previous experiment: Refining a LA posterior, even with a very simple and cheap NF, makes the weight-space posterior closer to that of HMC. Moreover, the calibration performance (in terms of NLL) also becomes better.

## C.3 Costs

We compare the theoretical costs of the refinement method in Table 9. For comparisons, we include the costs of popular Bayesian baselines: deep ensemble [20], SWAG [13], and MultiSWAG [43]. Our refinement method introduces overheads over the very cheap Kronecker-factored last-layer LA [6, Fig. 5, Tab. 5] due to the need of training the normalizing flow and feeding posterior samples forward through it during prediction. Combined with our findings in Section 6.4, our method only introduces a small overall overhead on top of the standard MAP network. Finally, using synthetic datasets, we show the empirical costs of performing refinement on different number of classes in Fig. 9. We note that even with 4096 features (e.g. in WideResNet-50-2) and 1000 classes (e.g. in ImageNet), the refinement method is cost-efficient.

## C.4 Weight-space distributions obtained by refinement

To validate that the refinement technique yields accurate posterior approximations, we plot the empirical marginal densities $q(w_i \mid \mathcal{D})$ in Figs. 10 to 12. We validate that the refinement method makes the crude, base LA posteriors closer to HMC in the weight space.

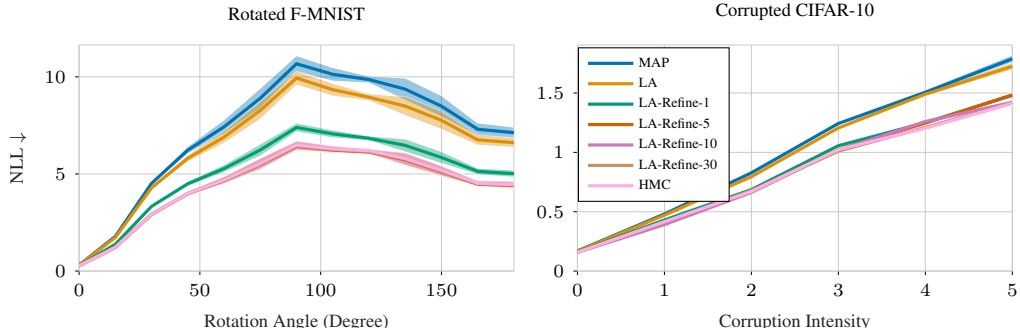

Figure 8: Calibration under dataset shifts in terms of NLL—lower is better.

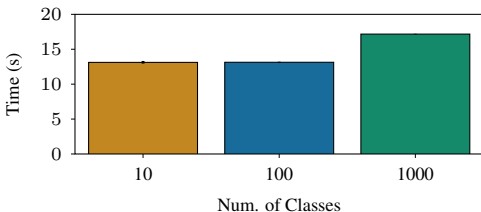

Figure 9: Empirical training costs of the refinement method, with $4096$ features—this simulates the upper end of commonly-used networks' last-layer features, e.g. WideResNet-50-2. The dataset is synthetically generated by drawing $10000$ points from a mixture of Gaussians, where number of modes equals number of classes. The training protocol for the NF is identical to the one used in the main text. A single RTX 2080Ti GPU is used to perform this experiment.

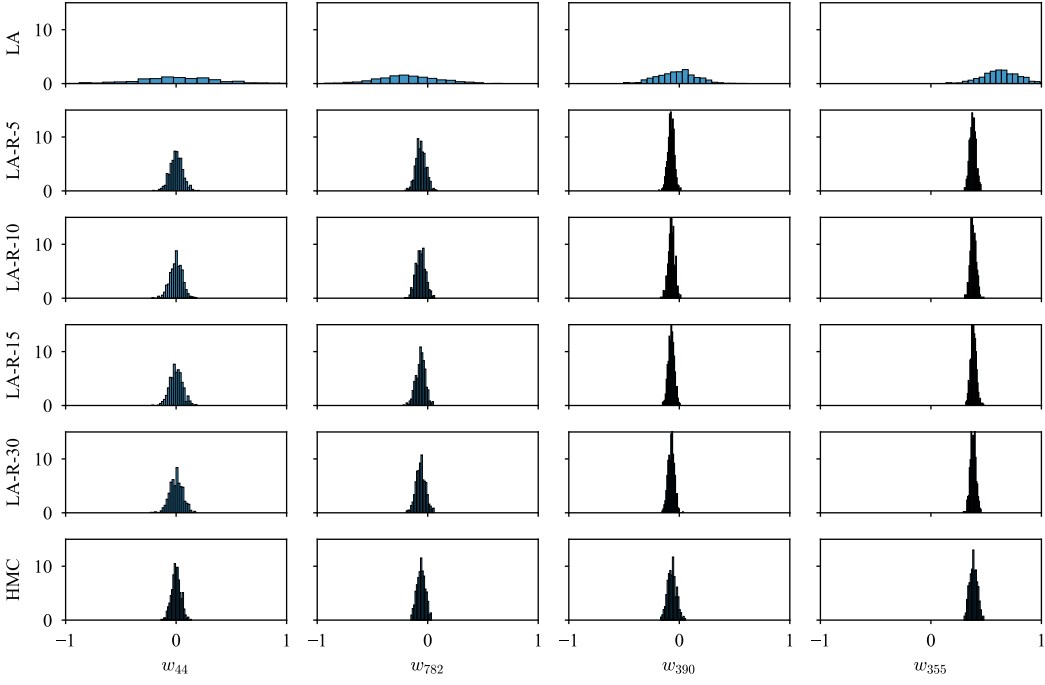

Figure 10: Empirical marginal posterior densities of some F-MNIST BNNs' random weights. "LA-R" is an abbreviation to "LA-Refine".

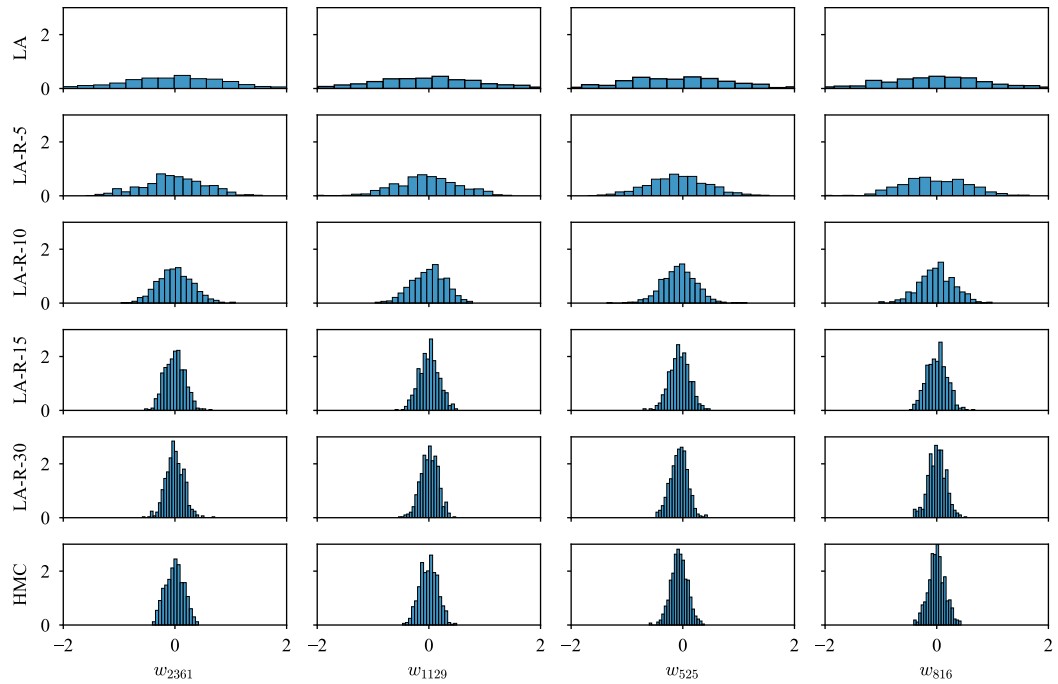

Figure 11: Empirical marginal posterior densities of some CIFAR-10 BNNs' random weights. "LA-R" is an abbreviation to "LA-Refine".

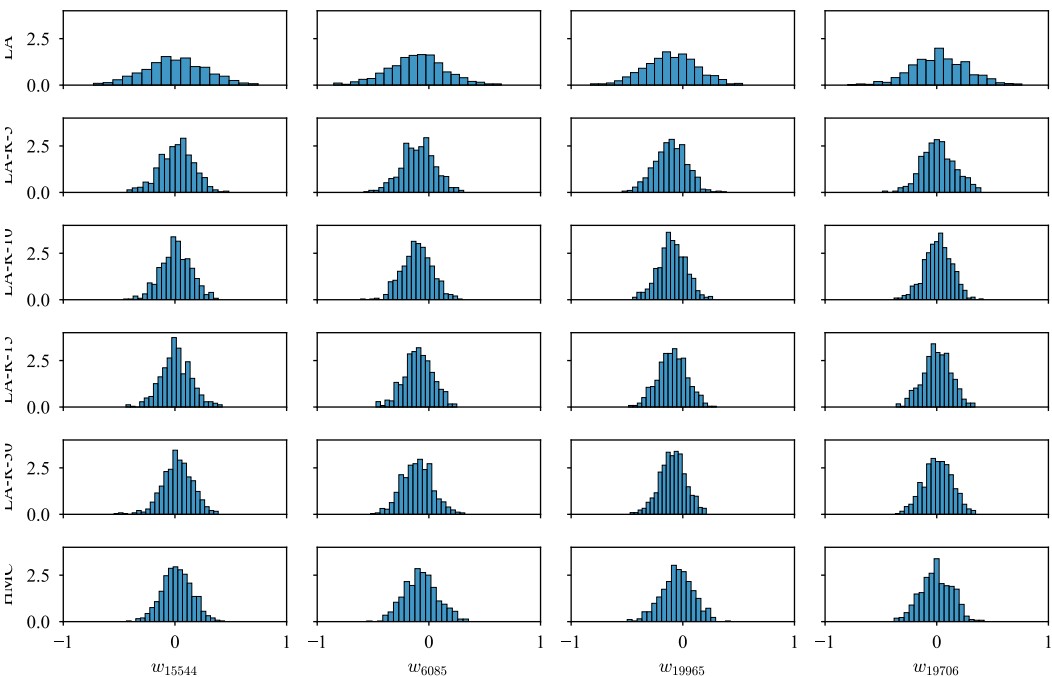

Figure 12: Empirical marginal posterior densities of some CIFAR-100 BNNs' random weights. "LA-R" is an abbreviation to "LA-Refine".

Table 7: Detailed OOD detection results. Values are FPR95. "LA-R" stands for "LA-Refine".

| Datasets | VB* | CSGHMC* | LA | LA-R-1 | LA-R-5 | LA-R-10 | LA-R-30 | HMC |
|---|---|---|---|---|---|---|---|---|
| **FMNIST** | | | | | | | | |
| EMNIST | 83.4±0.6 | 86.5±0.5 | 84.7±0.7 | 85.4±0.8 | 87.6±0.6 | 87.6±0.6 | 87.6±0.6 | 87.2±0.6 |
| MNIST | 76.0±1.6 | 75.8±1.8 | 77.9±0.8 | 77.5±0.9 | 79.6±1.0 | 79.6±1.0 | 79.6±1.1 | 79.0±0.9 |
| KMNIST | 71.3±0.9 | 74.4±0.5 | 78.5±0.6 | 78.3±0.8 | 79.9±0.9 | 79.9±0.9 | 79.9±0.9 | 79.4±0.9 |
| **CIFAR-10** | | | | | | | | |
| SVHN | 66.1±1.2 | 59.8±1.4 | 38.3±2.9 | 40.1±3.4 | 38.2±3.2 | 36.5±3.0 | 35.8±2.9 | 36.0±3.0 |
| LSUN | 53.3±2.5 | 51.7±1.5 | 51.1±1.1 | 46.9±0.5 | 46.7±0.6 | 46.9±0.7 | 47.1±0.5 | 46.7±0.8 |
| CIFAR-100 | 69.3±0.2 | 64.6±0.3 | 58.2±0.8 | 56.1±0.6 | 55.7±0.8 | 55.3±0.6 | 55.2±0.5 | 55.3±0.4 |
| **CIFAR-100** | | | | | | | | |
| SVHN | 81.7±0.7 | 75.9±1.5 | 82.2±0.8 | 77.7±1.2 | 78.1±1.3 | 77.9±1.5 | 78.3±1.6 | 78.2±1.6 |
| LSUN | 76.6±1.8 | 79.3±1.8 | 75.5±1.6 | 75.1±1.2 | 75.7±1.4 | 75.9±1.2 | 75.8±1.3 | 75.5±1.7 |
| CIFAR-10 | 84.2±0.4 | 82.8±0.3 | 81.0±0.3 | 79.1±0.4 | 79.5±0.4 | 79.5±0.4 | 79.6±0.4 | 79.7±0.2 |

Table 8: Text classification performance.

| | 20NG | | SST | | TREC | |
|---|---|---|---|---|---|---|
| **Methods** | **MMD ↓** | **NLL ↓** | **MMD ↓** | **NLL ↓** | **MMD ↓** | **NLL ↓** |
| MAP | 0.834 | 1.3038 | 0.515 | 1.3125 | 0.620 | 1.4289 |
| LA | 0.808 | 1.2772 | 0.474 | 1.2570 | 0.708 | 1.4206 |
| LA-Refine-5 | 0.643 | 0.9904 | 0.126 | 1.0464 | 0.554 | 1.3375 |
| HMC | - | 0.9838 | - | 1.0787 | - | 1.1545 |

Table 9: Theoretical cost of our method. $M$ is the number of parameters of the base neural network, $N$ is the number of training data, $C$ is the number of classes, $P$ is the number of last-layer features, $F$ is the length of the normalizing flow, $S$ is the number of MC samples for approximating the predictive distribution, $R$ is the number of model snapshots, $K$ is the number ensemble components.

| | | Prediction | |
|---|---|---|---|
| **Methods** | **Overhead over MAP** | **Computation** | **Memory** |
| MAP | - | $M$ | $M$ |
| Last-Layer LA | $NM + C^3 + P^3$ | $SCP$ | $M + C^2 + P^2$ |
| **Last-Layer LA+Refine** | $NM + C^3 + P^3 + NFCP$ | $SFCP$ | $M + C^2 + P^2 + FCP$ |
| SWAG | $RNM$ | $SRM$ | $RM$ |
| DE | - | $KM$ | $KM$ |
| MultiSWAG | $KRNM$ | $KSRM$ | $KRM$ |