# OpenReview forum: "Posterior Refinement Improves Sample Efficiency in Bayesian Neural Networks"
_NeurIPS.cc/2022/Conference — NeurIPS 2022 Accept_

### Official Review · Reviewer_qker · 2022-07-04

**Rating:** 7
**Confidence:** 4
**Soundness:** 4 excellent
**Presentation:** 4 excellent
**Contribution:** 3 good

**Summary:**

The paper addresses the problem of scalable Bayesian inference for large-scale neural networks. The core idea of the paper is to first do a (last layer) Laplace approximation of the posterior of the network weights and then refine it using normalizing flows. That is, the paper proposes to use a Laplace approximation as base distribution in normalizing flows instead of a fixed distribution, e.g. a standardized Gaussian.

The contribution of the paper is as follows:
- a discussion of shortcomings of common strategies for computing predictive distributions (by averaging over the approximate posterior). Specifically, they contrast linearization and probit approximations with Monte Carlo integration.
- propose to use a Laplace approximation as base distribution in a normalizing flows approximation
- a set of thorough numerical experiments to support the claims.


**Questions:**

- Can the explanation described in lines 190-200 be further justified/substantiated? For example by inspecting/visualizing the pairs of approximate posterior and approximate posterior predictive in the two cases?
- How much tuning was needed for optimizing the flows? Did you use the same optimization procedure/search parameters for all problems?

**Limitations:**

Yes.

**Strengths And Weaknesses:**

The paper proposes to combine two existing approximate inference techniques: Laplace approximations and normalizing flows. The idea is not revolutionary, but it is simple, effective, and original to the best of my knowledge. The paper is very well-structured, well-written and easy to follow. The claims of the paper are not justified theoretically, but they are supported by a set of high-quality numerical experiments.


There are a few things that could be slightly clarified:

- In my perspective, the quality of an approximate posterior predictive distribution depends on two things: 1) the quality of the posterior distribution of the model and 2) the accuracy of the integration method used for averaging over the posterior distribution. Therefore, using high-fidelity integration methods (e.g. MC integration with S -> infinity) will not save you if the quality of the posterior approximation is sufficiently poor. I think this is in line with the message of the paper, but it appears a bit convoluted in the discussion of the effect of the MC sample size on the accuracy of the predictive distribution (in my opinion)
- The difference between the top and bottom plot in Figure 5 could be emphasized visually and/or in the text. It took me a while to realize the difference.

---

> ### Author Response · Authors · 2022-08-01
> **Response**
>
> Thanks a lot for the review! We hope that our clarifications below can address your issues! If not, please do let us know in the discussion phase.
>
> - **Clarify the story?** Yes, you are correct that there are two things affecting the predictive quality, i.e. the quality of the weight-space approximation and the accuracy of the integration method. We will make this clearer in the next revision. Please refer also to our response to **_Reviewer 5mzJ_** for an additional perspective.
> - **Multiclass probit limitations? Make Fig. 5  more obvious?** The MPA only uses the diagonal elements of the covariance over the network outputs, among other approximations, which makes it underconfident. We have added the MPA’s derivation in Appendix A—which makes our point more formal— and made Fig. 5 clearer by changing the colormap.
> - **Refinement’s hyperparameter tuning?** We did not do extensive hyperparameter tuning for refinement: we use Adam with its default hyperparameters along with the cosine annealing for 20 epochs. We use the same setup for all problems.

---

### Official Review · Reviewer_5mzJ · 2022-07-11

**Rating:** 5
**Confidence:** 4
**Soundness:** 3 good
**Presentation:** 3 good
**Contribution:** 3 good

**Summary:**

The paper considers the problem of posterior approximation in Bayesian neural networks. First, the authors show that the accurate approximation with Monte Carlo requires a large number of samples, far exceeding the number of samples typically used by practitioners. Next, the authors show that (1) a large number of samples from a poor posterior approximation and (2) crude analytic approximations to the MC integral both lead to suboptimal performance in Bayesian neural networks. Finally, the authors suggest to use a variational refinement of the Laplace approximation with a normalizing flow to approximate the posterior in last layer BNNs, and show good performance.

**Questions:**

**Question 1**. Where do you expect the proposed method to be most useful? It appears that the primary strength is improving the uncertainty calibration for deep neural networks. How does it compare to specialized calibration methods, e.g. temperature scaling?

**Question 2**. How much is the proposed method affected by the size of the last layer, and the number of classes in particular? Is it cheap / computationally feasible to apply it to an ImageNet-scale model?

**Limitations:**

No issues

**Strengths And Weaknesses:**

**Strength 1**. The authors show many interesting observations, e.g. Figure 3 suggesting that 100 samples is insufficient for an MC approximation to even a very simple model, and Figure 4 showing the discrepancy between the MC approximation and analytic approximation to the posterior predictive.

**Strength 2**. The proposed approach is practical, as it is a cheap post-processing step to a pre-trained neural network. The empirical results are promising.

**Strength 3**. It is nice that the authors included MMD distance estimates in Table 3 and showed that the refined posterior is closer to the HMC posterior.

**Weakness 1**. The narrative of the paper is not very focused. In particular, it is not obvious, why the paper discusses the number of samples required for an MC approximation and the quality of analytic linearization-based approximations. The proposed method doesn't follow logically from these experiments, and the connection between these two parts of the paper is relatively weak.

**Weakness 2**. The empirical evaluation is not very strong. In terms of in-distribution performance, the authors only compare the proposed method to last-layer baselines, and achieve an improvement in uncertainty / NLL, but not accuracy (according to appendix Table 5). The results on out-of-distribution detection (appendix Table 7) show improvement over full parameter-space Bayesian methods, but do not include specialized out-of-distribution detection methods.

Overall, the paper shows interesting observations, and some promising results. However, the story feels disconnected, and the empirical observation doesn't prove that the proposed method will be useful for practitioners.

---

> ### Author Response · Authors · 2022-08-01
> **Response**
>
> Thanks for the review! We hope that our clarifications below can convince you more—please let us know if you have any additional questions!
>
> - **Clarify the narrative—it’s currently not very focused?** The story of the paper is as follows (see also **_Reviewer qker_** ’s review since they elegantly captured this)---we will clarify this in the next revision:
>   - We observe that there are two parts contributing to the inaccuracy in the predictive distribution of a BNNs: (i) the weight-space approximation and (ii) the integration method to do Bayesian model averaging.
>   - We show that, while the accuracy of the integration method is impactful (i.e. the number of samples in MC integration), it is less important than the quality of the posterior approximation itself (Table 1). I.e. higher-quality weight-space approximations yield samples that are more efficient—even small-sample MC integration can already yield good predictive distributions.
>   - A natural question would be whether alternatives to MC integration, such as linearization, can yield faithful predictive distributions even with inaccurate posteriors. The answer seems generally to be negative, except in a special case where Gauss-Newton Hessian approximation is used.
>   - It is thus instructive to use MC integration for general cases but under an accurate posterior approximation. However, accurate approximations such as HMC are very expensive. In this work, we provide a low-cost alternative.
> - **Lacking empirical evaluations?**
>   - Regarding other than last-layer baselines for in-distribution experiments, please refer to Daxberger et al., _Laplace redux—effortless Bayesian deep learning_, NeurIPS 2021. We will make this clearer in the next revision.
>   - Regarding accuracy, the main goal of the refinement method is to match HMC’s performance (which is considered the gold-standard baseline in literature). Notice that the accuracy of the base Laplace approximation itself is already on par with HMC—the refinement thus doesn’t gain any accuracy in this case.
>   - As for comparison against specialized OOD baselines, it is less relevant for BNNs since it has been shown that standard BNNs underperform against them, see Kristiadi et al., _Being a bit frequentist improves Bayesian neural networks_, AISTATS 2022. Note that, our refinement method is compatible with theirs.
> - **When is refinement useful (Compare to temperature scaling)?** You are correct that our method is best suited for obtaining good-quality predictive distributions. Compared to specialized, ad-hoc calibration methods like temperature scaling, our method should perform better due to the fact that it is a principled Bayesian method—see Kristiadi et al., _Being Bayesian even just a bit, fixes overconfidence in ReLU networks_, ICML 2020 for comparisons between (unrefined) Bayesian methods and temperature scaling for calibration and mitigating overconfidence.
> - **Costs vs. last-layer’s size? Scale to ImageNet models?** Using a synthetic dataset to allow for easily controlling the experiment’s parameters (number of classes, number of features, number of data points), we found that the cost of refinement (or the radial flow in general, as implemented in Pyro) is not very sensitive w.r.t. `n_features` or `n_classes`, for standard values that are often used in practice. (We tested up to the values of 4096 and 1000 for the former and the latter, respectively. This takes ~1.2s per epoch for 10k data points.) Combining this observation with Fig. 7 (bottom), we can thus conclude that, in practice, the length of the flow affect the cost more than the problem dimensionality. Fortunately, as we show in Fig. 7 (top), short flows are already sufficient to obtain good performance.

---

> > ### Comment · Reviewer_5mzJ · 2022-08-06
> > **Thank you for the rebuttal**
> >
> > Dear authors, thank you for the rebuttal!
> >
> > I understand the story of the paper as you described it. It makes sense, but also I still think that it is not particularly focused. The results on the number of MC samples required / linearization do not directly connect to the method directly. Regarding the empirical results, it sounds like the main hope is that the method will provide high-quality uncertainty estimates. I think this statement should be tested more carefully, with comparisons to targeted methods beyond last-layer BNNs. I don't think
> > > our method should perform better due to the fact that it is a principled Bayesian method
> >
> > is a sufficient justification, and I think explicit comparisons should be included.
> >
> > That being said, I am still leaning towards acceptance, so I maintain my score of 5 for now.

---

> > > ### Author Response · Authors · 2022-08-07
> > > **Thanks for the comment!**
> > >
> > > Thanks a lot for your further comments! We hope to clarify a couple of points:
> > >
> > > > The results on the number of MC samples required / linearization do not directly connect to the method directly.
> > >
> > > They are connected to the proposed refinement method directly: We showed that MC-predictive is desirable since it has a theoretical guarantee and its popular alternative, linearization, is not generally applicable and introduces biases. But, MC-predictive can be expensive due to the number of samples needed. We showed that the key to being efficient in obtaining MC-predictive is by having a fine-grained posterior, such as HMC. But, HMC is expensive and thus we propose the refinement framework, which is much more cost-efficient in obtaining accurate weight-space posteriors.
> > >
> > > >  I don't think "our method should perform better due to the fact that it is a principled Bayesian method" is a sufficient justification, and I think explicit comparisons should be included
> > >
> > > The quoted statement refers specifically to the finding in [1], esp. in their Fig. 1 and Fig. 4, where temperature scaling does not yield good calibration outside of the data region, unlike Bayesian methods. For explicit comparisons re. in-distribution calibration, please refer to [2] (Fig. 6 and Tab. 1)---Bayesian methods are better calibrated than temperature scaling in standard benchmark datasets. We will include similar explicit comparisons and discussions in the next revision. We will also include all-layer baselines, similar to [3] (Fig. 4).
> > >
> > >
> > >
> > > **References**
> > >
> > > [1] Kristiadi et al., Being Bayesian even just a bit, fixes overconfidence in ReLU networks, ICML 2020
> > >
> > > [2] Kristiadi et al., An Infinite-Feature Extension for Bayesian ReLU Nets That Fixes Their Asymptotic Overconfidence, NeurIPS 2021
> > >
> > > [3] Daxberger et al., Laplace Redux – Effortless Bayesian Deep Learning, NeurIPS 2021

---

### Official Review · Reviewer_4ug5 · 2022-07-17

**Rating:** 6
**Confidence:** 3
**Soundness:** 3 good
**Presentation:** 3 good
**Contribution:** 2 fair

**Summary:**

First, the authors of this paper propose an experimental analysis of failure modes of a given way of building Bayesian Neural Networks (Bayesian NNs, BNNs), namely, MC integration over samples from an approximate posterior.
Then, they propose a cheap and accurate method, which avoids these failure modes: replace the last layer of a NN by a variational layer, whose distribution is generated by a trained Normalizing Flow (which spans a much larger space of distributions than the commonly used product of Gaussians), then perform MC integration on top of that. They compare experimentally their method to the "gold standard" HMC.

## Failure modes of MC methods

The authors recall (and prove experimentally) that MC methods do not lead to good predictive performances if the underlying approximation of the Bayesian posterior is too rough. For instance: Laplace Approximation (LA) and Variational Bayes (VB) ; notably, these approximations of the posterior assume Gaussian and independent parameters.

One of the pitfalls is also the lack of exploration of the parameter space: deep ensembles perform way better, even with very few samples.

## Last-layer Bayesian NNs

The authors emphasize the idea of building "Bayesian" NNs only by replacing the usual last layer by a *Bayesian* last layer. Thus, the proposed methods relies heavily on "Laplace redux–effortless Bayesian deep learning" (Daxberger et al., 2021) [6]. Actually, the authors propose an improvement of [6]: instead of performing a simple LA on the last layer, the authors add Normalizing Flow (NF) to it, in order to increase the variety of reachable distributions.

**Questions:**

The phenomenon observed in [6] and by the authors of the current paper, i.e., the importance of the last layer when doing approximate Bayesian inference, is not understood:
 * According to the authors, which set of experiments could help us to understand it?
 * Is this a way for the NN to calibrate itself automatically, which would be impossible if the distribution of the final layer weights was constrained to Gaussians?
 * How would the NN react to other constrained distributions (Laplace, Weibull, etc.)? Is this related to the thickness of the tails?

The theoretical questions brought by the experiments done by the authors are a strength of the paper. These points should probably be discussed somewhere in the paper, with possibly some preliminary experiments.

**Limitations:**

-

**Strengths And Weaknesses:**

## Strengths

The authors motivate their approach by an analysis of failure of common methods. So, the proposed method is justified, at least experimentally.

## Weaknesses

The main weakness of this paper is its significance, compared to what is already known and used.

For instance, it is already known that sampling NNs from the same "basin of attraction" is not working very well compared to HMC and deep ensembles. So, the poor results of VB + MC or LA + MC in Section 3 are not surprising. For instance, see the exhaustive paper "Bayesian deep learning and a probabilistic perspective of generalization", Wilson and Izmailov, 2020.

About the proposed new method, this is a variation of one of the methods proposed in [6], which is actually an extensive study of the LA in many setups, with practical consideration and available code.

## Other

The experiments are limited to F-MNIST and CIFAR-10 and CIFAR-100. Since the results (tables 3, 4, 5) are clear and convincing on many aspects (calibration error, distance to the gold standard HMC, OOD detection), this should not be a problem.

---

> ### Author Response · Authors · 2022-08-01
> **Reponse**
>
> Thanks for your review! We hope that our response below can make you even more confident about our work. We will be very happy to discuss any follow-up questions you might have.
>
> - **Significance of the insights vis-à-vis e.g. Wilson & Izmailov?** Our contribution in Sec. 3 is: We study the effect of weight-space approximation quality against the number of samples needed to obtain good predictive distributions. (See also our responses to **_Reviewer qker_** and **_Reviewer 5mzJ_**.) While our results also reaffirm the widely held belief that single-basin approximations are inferior to more fine-grained counterparts, our conclusion differs from that of Wilson & Izmailov: They argue in their Sec. 3.2 that  “[...] Ultimately, the goal is to accurately compute the predictive distribution, rather than find a generally accurate representation of the posterior. [...]”. However, as we have shown in our work, finding an accurate weight-space posterior approximation is still a worthy goal. Note that we do so using only a single network, unlike deep ensemble or MultiSWAG, and thus our method is much cheaper.
> - **The refinement method, in comparison to Daxberger et al. [6]?** The proposed refinement method is generally applicable and not tied to specific posterior approximations (e.g. Laplace): The only requirement of our method is a parametric approximation to the posterior (e.g. Laplace, VB—not necessarily has to be Gaussian). However, when applied to Laplace, our method becomes even more compelling in practice due to the fact that Laplace is _post-hoc_ and thus the resulting refined-Laplace method is cheap.
> - **What helps to understand last-layer approximations?** We believe last-layer approximations are natural if one excludes full-layer ones. This is due to the role of the last layer which directly maps the feature space to the output space of a NN. So, given fixed feature vectors (obtained by freezing the previous layers and feedforward the training examples through them), one has a simple linear model which is easy to work with. As a big bonus, last-layer approximations are generally cheap since the dimensionality of the last layer of a NN tends to be smaller than other layers’. In terms of its performance, a straightforward experiment to validate the choice of last-layer approximations is by simply doing a Laplace approximation/variational inference on each layer separately and comparing the resulting performance on e.g. calibration. This can be done easily using recent libraries for LA/VI. However, while interesting, we believe that this experiment answers a problem that lies outside our paper’s specific topic.
> - **Is this a way to calibrate NN automatically?** Our goal is to match the performance given by the gold-standard HMC, cheaply. It is true that HMC often yields better-calibrated predictive distributions, and thus, by extension, the proposed refinement method can also make standard parametric approximations better calibrated. However, it is not true that our method is a foolproof nor automatic way to calibrate NNs. This is because even HMC itself is not foolproof—in particular, the prior hyperparameters and model misspecification nontrivially affect calibration quality.
> - **Non-Gaussian but still parametric approximation (e.g. VB with Laplace distributions)?** Heavier-tailed parametric approximations might be useful in some cases (e.g. when the basin of attraction locally also has heavy tails). However, a simple counterexample to this is already available in the text: Fig. 6 shows a true posterior that is skewed, so even with e.g. the Laplace distribution, one cannot obtain an accurate approximation. The refinement framework bypasses these prescribed, constrained approximations. I.e. by using refinement, one can flexibly fit complex posterior distributions—just like HMC—in a cost-efficient manner.

---

### Official Review · Reviewer_mJ2u · 2022-07-19

**Rating:** 6
**Confidence:** 2
**Soundness:** 2 fair
**Presentation:** 3 good
**Contribution:** 2 fair

**Summary:**

The paper studies empirically the current limitations of posterior estimation in Bayesian Neural Networks, focusing on Monte Carlo integrations as a simple golden standard and comparing its problems and limitations to the more recent techniques based on network linearization.
The authors then propose a post-hoc posterior refinement technique for posteriors to be estimated through MC integration and validate it empirically against the other analyzed techniques.

**Questions:**

1. Why Figure 5 should convince the reader that MPA is underconfident?

2. Why figure 2.c should convince the reader that linearization-based approaches are being overconfident? Is there a ground truth I'm missing?

3. Did you try different flows? What should convince the reader that this is a design choice not worth discussing or that it does not have a big impact on the final performance of the method?

**********---------------------**********

After the author's response I've increased my score to 6.

**Limitations:**

Big picture limitations were appropriately addressed.
However, I think the paper would greatly benefit from a more structured section or paragraph detailing the computational limitations, especially when compared to other baselines and related work. This doesn't need to be an empirical study of all methods computational costs, but just a clearer outline of the costs implied by each of the mentioned methods.

**Strengths And Weaknesses:**

Strengths
Overall, the paper is well written. The problem is clearly stated and so are the main approaches to it.
The proposed method is simple to understand and looks computationally cheap enough given the advantages it brings, and section 3 has some compelling arguments on why current methods should not be trusted.

Weaknesses
Section 3.3 feels confusing. It doesn't a good job in convincing the reader that linearization-based alternatives are bad enough to disregard a priori. Figure 5 failed in convincing me that the qualitative estimate of MC or MPA was either better or worse than the competitor.
Empirically, the range of experiments is satisfying, however the fact that the tested models are only a Le-Net and a WideResNet leaves the reader wondering how much the results would change with different type of architectures, e.g. even a simple Gated Recurrent Unit on Emotions or Newsgroup-20 or a perceptron on the same proposed datasets.

---

> ### Author Response · Authors · 2022-08-01
> **Reponse**
>
> Thanks for your review! We hope that our response below can convince you further. If you have any additional doubts, please let us know!
>
> - **Figure 5?** The MPA only uses the diagonal elements of the covariance over the network outputs, among other approximations, which makes it underconfident. We have added the MPA’s derivation in Appendix A—which makes our point more formal— and made Fig. 5 clearer by changing the colormap.
> - **Fig. 4c?** We don’t claim that linearization induces underconfident predictions per se: rather, we show that linearization, in general, induces biases compared to the ground-truth MC-integrated predictive (in our case, with $10000$ samples).
> - **Different flows for refinement?** Yes, we tried a different flow: in Fig. 6(c), we use the planar instead of radial flow. In any case, the radial flow is mainly used in the paper since it is among the simplest normalizing flows available—for a space of dimension $D$, it only has $D+2$ parameters on each layer (see Rezende & Mohamed, 2015, Sec. 4.1). Even then, we can already show that refinement is effective, even with short flow length (see Fig. 7). While more sophisticated, more expressive flows might improve this further, they come with higher computational costs and is harder to optimize/need to be tuned more.
> - **Elaborate more the cost limitation?** Please find below (Table 1) the theoretical costs of our method, in comparison to other Bayesian methods. Additionally, please refer to Fig. 5 of Daxberger et al., _Laplace redux—effortless Bayesian deep learning_, NeurIPS 2021 in conjunction to our Fig. 7 (bottom) for the concrete, practical values. Please refer also to the last point in our response to **_Reviewer 5mzJ_**.
>
> Table 1: Complexities of on top of MAP in O-notation. Radial flow is assumed for “Refine”.
>
> |  | Inference | Prediction (Computation) | Prediction (Memory) |
> |---|---|---|---|
> | MAP | - | M | M |
> | LLLA | NM + C^3 + P^3 | SM | M + C^2 + P^2 |
> | LLLA+Refine | NM + C^3 + P^3 + NFCP | SFCP | FCP |
> | SWAG | RNM | SRM | RM |
> | DE | - | KM | KM |
> | MultiSWAG | KRNM | KSRM | KRM |
>
> M=nr, of parameters of the DNN, N=nr. of training data points, C=nr. of classes, P=nr. of last-layer features, F=normalizing flow length, S=nr. of MC samples for predictive distribution, R=nr. of model snapshots, K=nr. ensemble components.
>
> - **Additional task (GRU + 20Newsgroups)?** Please find below preliminary results on a text classification task with a GRU network on the 20NG dataset. We will update the appendix in the next revision, including also further datasets (SST, TREC).
>
> | Method    | MMD   | Acc. | NLL    |
> |-----------|-------|------|--------|
> | MAP       | 0.834 | 72.8 | 1.3038 |
> | LA        | 0.808 | 72.7 | 1.2772 |
> | LA+Refine | 0.643 | 72.1 | 0.9904 |
> | HMC       | 0     | 72.6 | 0.9838 |

---

### Author Response · Authors · 2022-08-08
**Thank you to the reviewers**

We thank all reviewers for their comments and proposals to further clarify our work. We hope that all concerns were sufficiently addressed by our replies. If you feel like your concerns and questions were not addressed to your satisfaction, we would highly appreciate a follow-up comment. Since the discussion period ends soon, we would like to clarify any remaining issues as soon as possible. Thanks again for all your work!

---

### Meta-Review · Area_Chair_Wpy6 · 2022-08-22

**Recommendation:** Accept
**Confidence:** Certain

**Metareview:**

The paper proposes a method to refine Gaussian approximations of the posterior in Bayesian computations by using the normalizing flow. Such Gaussian approximations are usually cheap to obtain, via Laplace approximation or variational Bayes. The method proposed by the authors outperform standard MC approaches and is competitive with the most sophisticated ones (Hamiltonian MC), while cheaper.

The reviewers praised the experimental results. They also liked the nice explanations and illustrations of the failure of the standard MC approaches. Some remarks about the limited novelty of this discussion with respect to existing works (e.g. Wilson and Izmailov, 2020) were satisfactorily addressed by the authors during the discussion with the reviewers. Overall, the reviewers agreed that, while the writing of the paper could be improved in parts, the discussion and the method proposed in this paper are a nice contribution to Bayesian learning, and will be useful to the community. I will therefore recommend to accept the paper. I encourage the authors to take into account the comments of the reviewers (especially Rev. 5mzJ) when preparing the camera-ready version of the paper.

**Award:**

No

---

### Decision · Program_Chairs · 2022-09-14

Accept